# A standard transformer and attention with linear biases for molecular conformer generation

## Abstract

Sampling low-energy molecular conformations, spatial arrangements of atoms in a molecule, is a critical task for many different calculations performed in the drug discovery and optimization process. Numerous specialized equivariant networks have been designed to generate molecular conformations from 2D molecular graphs. Recently, non-equivariant transformer models have emerged as a viable alternative due to their capability to scale to improve generalization. However, the concern has been that non-equivariant models require a large model size to compensate the lack of equivariant bias. In this paper, we demonstrate that a well-chosen positional encoding effectively addresses these size limitations. A standard transformer model incorporating relative positional encoding for molecular graphs when scaled to 25 million parameters surpasses the current state-of-the-art non-equivariant base model with 64 million parameters on the GEOM-DRUGS benchmark. We implemented relative positional encoding as a negative attention bias that linearly increases with the shortest path distances between graph nodes at varying slopes for different attention heads, similar to ALiBi, a widely adopted relative positional encoding technique in the NLP domain. This architecture has the potential to serve as a foundation for a novel class of generative models for molecular conformations.

## 1. Introduction

Molecules can adopt a distribution of low-energy states such that the 3D structure of a molecule is best represented by an ensemble of energetically accessible conformations. The conformation distribution informs the possible properties of a molecule, so computing the conformations is of critical importance for drug discovery and optimization applications, including docking, 3D-QSAR, and physico-chemical property estimation (Hawkins, 2017). Therefore, the core problem of molecular conformation generation (MCG) is to sample the conformation distribution to create representative ensembles under limited resource constraints.

MCG is a rapidly developing and promising domain for generative artificial intelligence (AI). Until recently, the primary direction of the research was development of specialized message passing networks to generate molecular conformations and protein structures. These networks are based on architectures with equivariant bias that use pairwise features, such as distances between atoms, or invariant features derived from local equivariant frames to exploit an invariance of molecular structures to rigid transformations. Notable examples of equivariant models are EGNN (Satorras et al., 2021) and Torsional Diffusion (Jing et al., 2022). However, multiple studies demonstrated that non-equivariant models can outperform non-equivariant networks at MCG. One of the first non-equivariant model, DMCG (Zhu et al., 2022), led in benchmarks on molecular conformation generation for a relatively long time, and was succeeded by the Molecular Conformer Fields (MCF) model (Wang et al., 2024), which achieved groundbreaking root mean square deviation (RMSD) metrics on the GEOM-DRUGS benchmark. A major advantage of non-equivariant models is the ability to utilize domain-agnostic architectures such as the standard transformer or MLP-mixer (Tolstikhin et al., 2021) models. For instance, MCF processed molecular graphs using the PerceiverIO network (Jaegle et al., 2021), which can handle a wide variety of input modalities. A domain-agnostic approach is simple and allows reuse of a rich set of architectures, codebases and training infrastructure that have been developed for transformer models.

Despite the success of non-equivariant models, they still exhibit limitations in certain aspects of architecture and performance. MCF obtained SOTA molecule structure generation by using a very large model, by the standards of the field of small molecue structure generation, comprising 242M parameters. Also, while MCF performed well on recall metrics, it showed modest precision results that were subsequently surpassed by a recent flow-matching model, ET-Flow, utlizing an equivariant transformer and harmonic diffusion prior (Hassan et al., 2024). In this paper, we aim

to improve on the performance of non-equivariant models at smaller model sizes to address the shortcoming of requiring large model size. To accomplish this, we designed a new non-equivariant diffusion model using a more advanced positional encoding (PE) compared to SOTA non-equivariant models.

The MCF model used graph Laplacian eigenvectors to represent graph node PE that were fed into PerceiverIO. In standard graph regression benchmarks, a graph transformer with a similar PE (Dwivedi & Bresson, 2021) has been surpassed by SAN (Kreuzer et al., 2021) and Graphormer (Ying et al., 2021). SAN uses a learnable PE from eigenvalues and eigenvectors, and Graphormer uses relative PE represented by a learnable attention bias indexed by the shortest path distances. Numerous other options exist for encoding graph structures in transformers; for a review, refer to (Rampášek et al., 2022). It is reasonable to assume that the performance of the non-equivariant model can be improved by using a more advanced PE. Indeed, as we show in this paper, incorporating a simple relative PE with a bias term in self-attention allowed us to create a SOTA model with a small number of parameters that competes with both MCF and ET-Flow models.

We further demonstrate that the difference seen in precision metrics between MCF and ET-Flow models is likely due to different treatment of stereochemistry information, specifically molecular chirality. Chirality tags were directly used as input features when training ET-Flow, but were absent in the MCF model. ET-Flow also used an additional chirality correction step to further improve metric performance. Here, we demonstrate that adding a chirality correction step to a non-equivariant model reduces the deficit of non-equivariant models in precision performance.

To train the model on a limited computational budget, we developed a two-stage training process. At the first stage, the model is trained on molecules with removed hydrogen atoms, which represent on average $\approx 44\%$ of the atoms in molecules in the GEOM-DRUGS dataset. At the second stage the model is finetuned on non-modified molecules (i.e. with hydrogens re-introduced). This training protocol accelerates model training by reducing the number of tokens used to represent molecules in the first stage of training, and its successful application unlocks opportunities for constructing new cascade diffusion models (Ho et al., 2022) for molecular conformations. In addition, we explored other aspects of non-equivariant diffusion models that have not attracted attention in the literature before. Most non-equivariant models employ raw atom coordinates in their feature set. As shown in other domains (Gorishniy et al., 2022), small MLP and sinusoidal continuous value encoders can enhance model convergence and improve model performance. Therefore, we investigated if

continuous value encoders could additionally improve the quality of generated molecular conformations.

Our key contributions can be summarized as follows

1. We developed a new transformer model for MCG with new relative positional encoding that demonstrates state-of-the art recall results for small and medium size models on the GEOM-DRUGS benchmark.

2. We highlight the importance of including/excluding chirality information to make fair comparisons between models and demonstrate that chirality correction for non-equivariant transformers can improve precision metrics.

3. Our simple two stage training protocol (removing/restoring hydrogen atoms) makes more efficient use of a limited computational budget when scaling models.

## 2. Related Work

In this section we discuss the alternative methods to AI for MCG, briefly review ML models for MCG, and discuss the use of positional encoding in graph transformers. Finally, we provide a detailed overview of GEOM dataset benchmarks and how they have been used in MCG models from the literature.

### 2.1. MCG methods

Conformer ensemble generation methods are usually classified as either stochastic or systematic, based on the type of search used (Hawkins, 2017). The most complex stochastic methods are molecular dynamics (MD) simulations, which simulate atoms using force-fields and are computationally expensive (Riniker & Landrum, 2015). Enhanced sampling methods, such as metadynamics, can help MD simulations explore more efficiently. The CREST software (Pracht et al., 2020; 2024) provides a suite that includes MD and metadynamics, and has been used to generate reference conformer ensembles in small molecule datasets such as GEOM (Axelrod & Gómez-Bombarelli, 2022) that are used as targets to benchmark conformer generation approaches. Alternative stochastic methods can improve speed by searching a lower-dimensional space than MD, using Monte Carlo search or genetic algorithms (GA). These methods can be subject to bias introduced by seed coordinates (Hawkins, 2017). The distance geometry (DG) approach (Havel et al., 1983) is used by many stochastic algorithms and helps to avoid seed bias by initializing coordinates based on distance constraints, and guides search towards conformations that match these predicted distances. Systematic, or rule-based, approaches typically divide the molecule into fragments and apply rules for

how each section should conform, using tools like fragment databases or torsion dictionaries, then reassemble the molecule (Hawkins, 2017). Popular rule-based approaches include OMEGA (Hawkins et al., 2010) and Frog2 (Miteva et al., 2010). While purely rule-based approaches are fast, reliance on fragment templates creates difficulties for large or flexible molecules (Ganea et al., 2021). Successful methods often combine multiple types of algorithm, for example, Balloon (Vainio & Johnson, 2007) initializes GAs using DG, and ETKDG (Riniker & Landrum, 2015) augments the popular cheminformatics software package RD-Kit implementation of DG generation (ETDG) with fragment knowledge. A new avenue for MCG comes from deep learning, and recently diffusion models (Ho et al., 2020) were adapted to generate distributions of conformers (Xu et al., 2022). Subsequently, diffusion models were able to outperform reference stochastic (ETKDG) and rule-based (OMEGA) methods at recreating CREST ensembles (Jing et al., 2022).

AI models for conformer generation can be categorized into models for direct MCG and models that produce spatial features of molecular graph such as inter-atomic distances, which are subsequently used to reconstruct atom coordinates with Euclidean distance geometry algorithms (Simm & Hernandez-Lobato, 2020). Numerous other graph models were developed for predictive and generative molecular analysis; for a recent review, refer to (Duval et al., 2023). The majority of these models operate in the Euclidean space of atom coordinates. An exception is Torsional Diffusion (Jing et al., 2022), which operates on spaces of torsional angles that have a much lower dimensionality. High numbers of iterations during inference became a restrictive factor for diffusion models. One of the solutions to the problem is harmonic diffusion (Jing et al., 2023), which initially generates distributions of molecule-like objects with bond lengths close to real molecules and can then be used as a prior for diffusion models. ET-Flow (Hassan et al., 2024), a flow-matching model trained on such a harmonic prior, requires only 50 iterations over the network during the inference stage.

A significant paradigm shift has occurred due to a recent publication. Instead of specialized graph neural networks, MCF (Wang et al., 2024) employed a universal transformer Perceiver IO (Jaegle et al., 2021) designed with predictive capabilities for multiple modalities. Utilizing standard architectures helps avoid model tuning and extensive hyperparameter searches while enhancing performance and scalability. The experiments in the MCF paper highlighted an important finding: sample quality improves with increasing model size. This discovery opened possibilities to enhance MCG with standard AI techniques, first by scaling a model to large sizes, then reducing the model size through methods such as quantization and distillation.

## 2.2. Positional encoding in graph transformers

The MCF model uses global PE, where each atom is assigned a vector comprising coordinates of the first $k$ non-trivial eigenvectors in the eigenbasis of the molecular graph's normalized Laplacian, ordered by ascending eigenvalues. This choice is likely dictated by the MCF problem statement to formulate the model as a diffusion probabilistic field (Zhuang et al., 2023). However, the literature around PE has shown development of more advanced PE schemes (Rampášek et al., 2022). We considered and tested multiple options, including transformers combined with GNNs (Min et al., 2022), ultimately selecting a relative PE approach.

Popular relative positional encoding in language models are RoPE (Su et al., 2024) and ALiBi (Press et al., 2021). In graphs, relative positional encoding can be represented by a bias term in attention that is calculated from some distance between nodes. In Graphormer (Ying et al., 2021), learnable biases $b_{\phi(v_i, v_j)}$ are employed, indexed by shortest path distance $\phi(v_i, v_j)$ between graph nodes $v_i$ and $v_j$,

$$A_{ij} = \frac{(\mathbf{h}_i \mathbf{W}_Q)(\mathbf{h}_j \mathbf{W}_K)^T}{\sqrt{d}} + b_{\phi(v_i, v_j)}, \qquad (1)$$

where $A_{ij}$ is $(i, j)$ - element of the Query-Key product matrix $\mathbf{A}$, and $\mathbf{h} \in \mathbb{R}^{1 \times d}$ is a transformer hidden state of dimension $d$. The mechanism of the learnable bias is attractive, as it allows bias variation across different transformer heads. For MCG, heads with a large bias can focus on encoding the graph, while heads with a minimal bias can primarily process atom coordinates.

## 2.3. Datasets and benchmarks for conformation prediction

The GEOM datasets (Axelrod & Gómez-Bombarelli, 2022) have been adopted for evaluation and comparison of machine learning methods for conformer generation. GEOM-DRUGS consists of 304,466 mid-sized molecules (averaging 44.4 atoms per molecule), and the GEOM-QM9 consists of 133,258 small molecules (averaging 18.0 atoms per molecule). Each data entry consists of a SMILES string and an ensemble of 3D conformers generated by simulation using the CREST method (Pracht et al., 2020; 2024). When training on these datasets, the goal is to train a model that reproduces the distribution of 3D conformations for each SMILES (or 2D graph). To evaluate model-generated conformations and compare distributions, RMSD is calculated between all reference and generated conformations for a molecule, from which two metrics are computed, coverage (COV) and absolute mean RMSD (AMR) (Zhou et al.,

2023), according to

$$\text{RMSD}(R, \bar{R}) = \min_{\Phi} \left( \frac{1}{n} \sum_{i=1}^{n} ||\Phi(R_i) - \bar{R}_i||^2 \right) \quad (2)$$

$$\text{COV}(S_g, S_r) = \frac{|\{R \in S_r | \text{RMSD}(R, \bar{R}) < \delta, \bar{R} \in S_g\}|}{|S_r|}$$

$$\tag{3}$$

$$\text{AMR}(S_g, S_r) = \frac{1}{|S_r|} \sum_{R \in S_r} \min_{\bar{R} \in S_g} \text{RMSD}(R, \bar{R}), \quad (4)$$

where $S_g$ and $S_r$ are the set of generated and ground-truth conformations for a molecule, respectively; $R$ and $\bar{R}$ are matrices of reference and generated conformations with the heavy atom coordinates in matrix rows, and $\Phi$ are the rigid transformations to align the generated conformation with the reference. In addition to the above recall metrics, precision metrics are usually reported, wherein $S_g$ and $S_r$ switch places in (3)-(4). In all preceding studies, these metrics are calculated on $2k$ generated conformers for each molecule, where $k$ is the number of conformers for a molecule in the test set.

Early SOTA on reproducing GEOM conformations was set by CGCF (Xu et al., 2021), which trained on only 50,000 conformations from each dataset and evaluated against conformations from 100 molecules. This work established the coverage threshold $\delta$ in (3) as 0.5Å and 1.25Å for GEOM-QM9 and GEOM-DRUGS, respectively. Subsequently, ConfGF (Shi et al., 2021) established SOTA after training with 5 conformations from each of 40,000 molecules from both GEOM datasets (200,000 conformations in training sets), and a test set of 200 molecules with between 50 and 500 conformations (22,408 total conformations for GEOM-QM9 and 14,324 for GEOM-DRUGS). Both GeoDiff (Xu et al., 2022) and DMCG (Zhu et al., 2022) used these training and test sets, which the latter termed "small-scale GEOM-DRUGS" and also trained on a larger set of 2,000,000 conformations from GEOM-DRUGS, furthering the benchmark SOTA. An earlier study that considered every molecule in the GEOM-Drugs dataset was GeoMol (Ganea et al., 2021), which established dataset indices for train/validation/test splits that have been reused in subsequent studies. The splits used 80%/10%/10% of the data, with the test split further downsampled to 1000 random molecules, creating splits containing 106,586/13,323/1,000 and 243,473/30,433/1,000 molecules for train/validation/test for GEOM-QM9 and GEOM-DRUGS, respectively. During training, a random sample of 10 and 20 conformations per molecules were used, for GEOM-QM9 and GEOM-DRUGS, respectively, while every conformer is used from the test split molecules during evaluation.

With Torsional Diffusion (Jing et al., 2022), a quantitative leap in the GEOM-DRUGS benchmark was achieved, such that the coverage threshold $\delta$ in (3) had to be reduced from 1.25Å to 0.75Å to observe differences between models. This and subsequent studies MCF (Wang et al., 2024), which established current SOTA for recall metrics, and ET-Flow (Hassan et al., 2024), which established current SOTA for precision metrics, all used the GeoMol data splits. However, comparisons between methods are still confounded as Torsional Diffusion and ET-Flow trained with 30 conformers per molecule for both GEOM-DRUGS and GEOM-QM9, while MCF used the 20 and 10 conformers for GEOM-DRUGS and GEOM-QM9, respectively, established by GeoMol.

The process for generating reference conformers in the GEOM datasets involves a graph re-identification step (Axelrod & Gómez-Bombarelli, 2022) that can assign different graphs to the same molecule, which are grouped according to the original SMILES. Therefore, RMSDs cannot necessarily be calculated between all reference and generated conformations for a molecule in the dataset. Torsional diffusion (Jing et al., 2022) simply dropped any such molecules from data splits. MCF (Wang et al., 2024) generated 2 conformations for each reference conformation, but, because RMSD can only be computed for conformers with identical graphs, some molecules effectively have a reduced number of RMSD values for computing COV and AMR, altering the metrics and confounding comparisons unless identical methods for handling these data discrepancies are used. Here we followed the MCF approach, keeping conformers grouped by reference molecule.

### 2.4. Molecule chirality

In the GEOM dataset all conformers of a given molecule have the same chirality. Thus, for optimal performance it may be important to include chirality in the feature set of 2D molecular graphs, as was done in Torsional Diffusion and ET-FLow. Torsional Diffusion generated conformations with the correct chirality by design, and ET-Flow used chirality tags in the atom feature set. Surprisingly, it is common for models in the literature to use 2D graph input without chirality features. For instance, the ensemble of conformers generated by MCF for a given molecule has a distribution of chiralities due to the lack of chirality information in the 2D graph feature set. In the GEOM-DRUGS test split 20% of molecules have one chiral center, and 8% of molecules have at least two chiral centers. Conformations generated with incorrect chirality will lower precision metrics, and could be one of the factors for the lower precision reported for MCF relative to ET-Flow. In this work, we did not fully explore this hypothesis and did not use chirality information in 2D graph input, as our main goal was to compare against the MCF architecture. However, we demonstrate here that a simplistic *post hoc* chirality correction can significantly improve model performance.

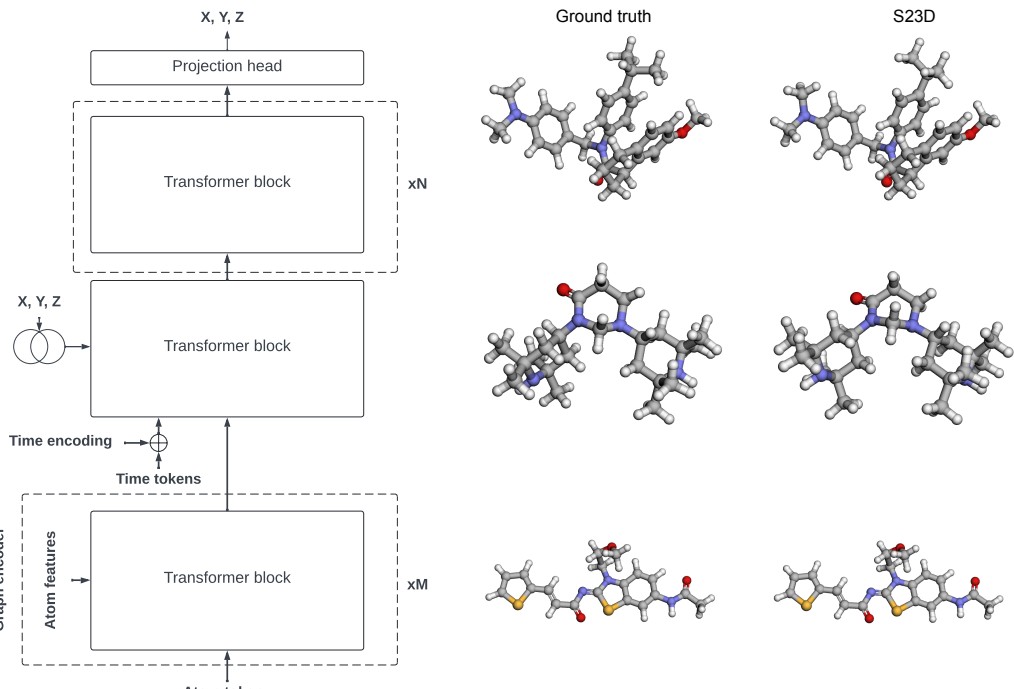

*Figure 1.* **Left**: S23D score network architecture. The backbone of the model is a standard transformer network. The first transformer blocks encode the 2D graph and atom features. Coordinate encodings are projected into hidden states of the transformer blocks in the second structural encoding blocks, where time tokens are also introduced. The head of the transformer is a projection layer that transforms each atom embedding into $\mathbb{R}^3$. **Right**: Example generated conformations. Ground truth conformations from GEOM-DRUGS test set (left) and aligned samples generated by the model (right).

## 3. Methods

### 3.1. Model architecture

In this paper, we used a diffusion model (Variance Preserving SDE (Song et al., 2020b), for equations see A.2) with a transformer score network. The overall structure of the score network is shown in Figure 1. The score network is composed of two subnets. The first subnetwork encodes 2D graph structure and transforms atom element embeddings into graph atom representations. The output of this network does not depend on atom coordinates and diffusion time. The output of the graph encoder is fed to a structural subnetwork that also receives 3D coordinates as input, and produces the score. Both subnetworks incorporate graph bias in the attention mechanism, as discussed below. The introduction of the graph encoder allows faster inference at larger model sizes, as the graph encoder only need to be run once during the inference stage, rather than on each diffusion step. The backbone of the model is the transformer block from the LLaMA architecture (Touvron et al., 2023). The only modifications to the LLaMa transformer block were: 1) injection of atom features into hidden states after the first normalization layer; and 2) linear attention bias calculated from shortest path distances. The final layer of the network is a linear layer for projection of atom embed-

dings into $\mathbb{R}^3$. Below, we describe components of the augmented transformer architecture and underlying motivation in detail.

#### 3.1.1. 2D MOLECULAR GRAPH ENCODING

Input tokens for the transformer were atom tokens. For atom tokenization, we selected chemical elements with frequent occurrences (B, Bi, Br, C, Cl, F, H, I, N, O, P, S, Si) and assigned a unique token embedding for each. For all other chemical elements, the embedding of the special **MASK** token was used.

The LLaMA transformer block contains normalization, attention, second normalization, and MLP layers. In the graph encoder, molecular graph features $f_i$ for an atom $i$ were injected between the first normalization and attention layers by projecting features into atom hidden states. In addition, similar to (Ma et al., 2023; Corso et al., 2020), we introduced an adaptive degree scaler to inject graph node degree $deg_i$ into a hidden state $\mathbf{h}_i \in \mathbb{R}^{1 \times d}$ for an atom $i$,

$$\mathbf{h}_i^{out} = \mathbf{h}_i \odot \boldsymbol{\theta}_1 + \mathbf{h}_i \odot \boldsymbol{\theta}_2 \cdot log(1 + deg_i) + \mathbf{f}_i \mathbf{W}_f, \quad (5)$$

where $\boldsymbol{\theta}_1$ and $\boldsymbol{\theta}_2 \in \mathbb{R}^{1 \times d}$ are learnable weights. The binary feature vector $\mathbf{f}_i \in \mathbb{R}^{1 \times 5}$ projected by the weight $\mathbf{W}_f \in \mathbb{R}^{5 \times d}$ consists of bond type and atom charge. Three

entries in the feature vector corresponds to distinct bond types (double, triple, aromatic), and a value of 1 means the atom is connected to at least one other atom with a bond of that type, while 0 means it is not. Two entries in $\mathbf{f}_i$ are allocated to a one-hot representation of positive, zero, and negative atom charges. We did not use chirality features as the main goal of the paper was to make a fair comparison between our and MCF models.

Implementation of the learnable bias in (1) requires querying lookup tables, which although highly optimized is still not an efficient operation on GPUs. It is also unclear how a model with learnable bias trained on certain graph sizes generalizes to larger molecular graphs. Therefore, to incorporate graph structure, we used a modification of (1) in all multi-head attention layers,

$$A_{ij}^{head} = \frac{(\mathbf{h}_i \mathbf{W}_Q^{head})(\mathbf{h}_j \mathbf{W}_K^{head})^T}{\sqrt{d^{head}}} - m_{head}\phi(v_i, v_j),$$
(6)

where $head$ specifies the Query-Key product matrix $\mathbf{A}$ and weights $\mathbf{W}_Q^{head}$ and $\mathbf{W}_K^{head}$ for a specific attention head, and $m_{head}$ is a head-specific slope. We used slopes that were used in the ALiBi positional encoding (Press et al., 2021), a geometric sequence that starts with $2^{\frac{-8}{n}}$ for $n$ attention heads and has the same value for the ratio. Since we had a mixture of time tokens and atom tokens in the structural subnetwork, zero bias with atoms tokens was applied for time tokens.

### 3.1.2. Spatial feature encoding

For 2D graph features, atom $i$ coordinate encodings $\mathbf{s}_i$ were projected into hidden states of atoms after the first normalization layer in the first transformer block of the second subnetwork,

$$\mathbf{h}_i^{out} = \mathbf{h}_i + \mathbf{s}_i \mathbf{W}_c.$$
(7)

In this work, we tested multiple types of continuous encoders modified from those proposed previously (Gorishniy et al., 2022) to obtain $\mathbf{s}_i$: 1) a plain coordinate encoding; 2) a sinusoidal encoding combined with atom coordinates; and 3) an MLP encoding:

1. $\mathbf{s}_i = \mathbf{r}_i = [x_1, x_2, x_3]$, where $x_i$ are plain normalized atom coordinates.

2. In the second encoder, atom coordinates $r_i$ were complemented with their sinusoidal encoding,

$$\begin{aligned} m_{ijc} &= sin(w_{jc}x_c) \\ l_{ijc} &= cos(\bar{w}_{jc}x_c) \end{aligned}$$
(8)

where $w_{j,c}$ and $\bar{w}_{jc}$ are trainable model weights initialized with $2^j\pi$, $j = 1, \ldots, 6$.

3. In the last encoder, $\mathbf{r}_i$ is transformed into $\mathbf{s}_i$ using two layer MLP with 128 hidden size and ReLU non-linearity.

### 3.1.3. Data augmentation

Random orthogonal transformations with matrices drawn from the O(3) Haar distribution (using class ortho_group from SciPy) were applied to all conformations. Note that non-equivariant models trained on GEOM-DRUGS without chirality features produce samples with random chirality even if augmentation transformations are from SO(3) group, and we decided to use O(3) group augmentation with *post hoc* chirality correction. Coordinate inputs to the transformer were translated by a random shift with components drawn from a uniform distribution with translations up to 30Å.

### 3.1.4. Chirality correction

We correct chirality with mirror reflection of conformations if Cahn-Ingold-Prelog labels produced by RDKit did not match the labels of the target molecule. Note, this operation corrects chirality of all conformers with single chiral centers, but rarely succeeds for conformers with multiple chiral centers. Thus, we expected a modest improvement in precision metrics, but did not expect to reach the precision performance of ET-Flow without further handling of chirality in the model.

## 4. Experiments

We empirically evaluated our model by comparing generated conformers with reference conformers given an input SMILES. We used metrics based on the RMSD between generated and reference conformers, as described in section 2.3. We evaluated using GEOM datasets, reporting results from GEOM-DRUGS as our primary experiment in section 4.2, with further breakdown of performance in the Appendix in section B.1.3. Also in the Appendix, we report benchmark results on GEOM-QM9 and GEOM-XL in sections B.1.1 and B.1.2, respectively, compare inference sampling time with MCF in section B.2.1 and report performance with different numbers of inference steps in section B.2.2.

### 4.1. Experimental Setup

Our model, which we termed 'S23D' ("SMILES to 3D"), was trained with two primary configurations of different sizes, 'small' (S23D-S, 8.6M parameters) and 'base' (S23D-B, 24.8M parameters). Parameters of these two primary models are described in section A.1. We trained several variants of S23D-S with different options. We tested the effect of training with different numbers of transformer

*Table 1.* Molecule conformer generation results on GEOM-DRUGS ($\delta = 0.75$Å). Sections of the table are seperated by approximate size of the model. Blue shows SOTA results within a model size range if the metric is only improved upon by a larger model, and **bold** shows overall SOTA results. 'Size' is model size in millions of parameters. '$K_G$' is number of conformers for each molecule in test split used during training. 'Chirality' indicates whether chirality information was used, either as features during training or corrected during generation. S23D-S and S23D-B indicate our small and base models respectively. '-M/N' show numbers of layers in each model section (see section A.1). (M), (S), and (C) show the coordinate encoding scheme used: MLP, sinusoidal, or plain coordinates, respectively (see section 3.1.2)

| | | | | RECALL | | | | PRECISION | | | |
| | | | | COVERAGE ↑ | | AMR ↓ | | COVERAGE ↑ | | AMR ↓ | |
| | SIZE (M) | $K_G$ | CHIRALITY | MEAN | MEDIAN | MEAN | MEDIAN | MEAN | MEDIAN | MEAN | MEDIAN |
|---|---|---|---|---|---|---|---|---|---|---|---|
| GEOMOL | | 20 | YES | 44.6 | 41.4 | 0.875 | 0.834 | 43.0 | 36.4 | 0.928 | 0.841 |
| TOR. DIFF. | | 30 | YES | 72.7 | 80.0 | 0.582 | 0.565 | 55.2 | 56.9 | 0.778 | 0.729 |
| MCF-S | 13 | 20 | NO | 79.4 | 87.5 | 0.512 | 0.492 | 57.4 | 57.6 | 0.761 | 0.715 |
| ET-FLOW - SS | 8.3 | 30 | YES | 79.6 | 84.6 | 0.439 | 0.406 | **75.2** | **81.7** | **0.517** | **0.442** |
| ET-FLOW - *SO(3)* | 9.1 | 30 | YES | 78.2 | 83.3 | 0.480 | 0.459 | 67.3 | 71.2 | 0.637 | 0.567 |
| S23D-S-1/9 (S) | 8.6 | 20 | NO | 80.7 | 89.9 | 0.483 | 0.458 | 57.5 | 57.5 | 0.757 | 0.700 |
| S23D-S-4/6 (S) | 8.6 | 20 | NO | 81.5 | 89.4 | 0.475 | 0.455 | 59.6 | 61.4 | 0.732 | 0.677 |
| S23D-S-1/9 (M) | 8.6 | 20 | NO | 81.5 | 90.0 | 0.473 | 0.444 | 58.1 | 57.8 | 0.751 | 0.702 |
| S23D-S-1/9 (C) | 8.6 | 20 | NO | 81.9 | 88.9 | 0.462 | 0.442 | 58.8 | 58.4 | 0.740 | 0.697 |
| S23D-S-1/9 (S) | 8.6 | 30 | NO | 81.5 | 89.6 | 0.475 | 0.451 | 58.4 | 58.9 | 0.749 | 0.692 |
| S23D-S-1/9 (S) | 8.6 | 20 | YES | 83.3 | 91.9 | 0.457 | 0.432 | 63.4 | 67.0 | 0.689 | 0.630 |
| S23D-S-1/9 (S) | 8.6 | 30 | YES | 83.8 | 92.0 | 0.449 | 0.417 | 64.1 | 67.5 | 0.680 | 0.622 |
| MCF-B | 64 | 20 | NO | 84.0 | 91.5 | 0.427 | 0.402 | 64.0 | 66.8 | 0.667 | 0.605 |
| S23D-B-1/13 (S) | 25 | 20 | NO | 84.5 | 91.7 | 0.420 | 0.387 | 62.5 | 63.4 | 0.690 | 0.626 |
| S23D-B-1/13 (S) | 25 | 20 | YES | **86.5** | **93.6** | 0.391 | 0.363 | 69.2 | 74.6 | 0.608 | 0.539 |
| MCF-L | 242 | 20 | NO | 84.7 | 92.2 | **0.390** | **0.247** | 66.8 | 71.3 | 0.618 | 0.530 |

blocks for the graph encoder and structural subnetwork sections of the model, while maintaining the same total model size, which we labeled with '-M/N', respectively (see section A.1, e.g. S23D-S-1/9 indicates 1 graph encoder block and 9 structural blocks). We also tested different coordinate encodings, which we label 'M' (MLP), 'S' (sinusoidal), or 'C' (plain coordinates) (see section 3.1.2, e.g. S23D-S-1/9 (S) indicates sinusoidal encoding was used).

To enable direct comparison with MCF, we trained models on the GEOM-DRUGS dataset with splits as in (Ganea et al., 2021), which are also used by Torsional Diffusion (Jing et al., 2022), MCF (Wang et al., 2024) and ET-Flow (Hassan et al., 2024). All experiments were conducted in two stages, first training for 100 epochs with hydrogen atoms removed, and second training for a further 25-35 epochs on the same data using complete molecules. MCF was trained for $750,000$ steps using batch size $512$, which is equivalent to $93.6$ epochs in our experiments. At the start of the second stage the embedding vector for hydrogen was initialized with the pre-trained embedding of the **MASK** token. During both training stages, we masked 10% of atoms with the token **MASK** for 1% of the molecules, selecting atoms to mask randomly with a probability inversely proportional to their frequencies in a given molecule.

Our primary comparisons were against MCF models, the current SOTA non-equivariant model, of similar size to our models (i.e. MCF-S and MCF-B), so we initially used 20

conformations per molecule and no chirality correction to match the experimental setup in (Wang et al., 2024). To understand how the number of conformations per molecule affects model performance, we also trained with 30 conformations per molecule, as used in (Jing et al., 2022; Wang et al., 2024), reducing the number of training epochs to 80 to offset the increased size of the training data. To provide a good comparison with ET-Flow, the current SOTA equivariant model, we trained with both 30 conformations per molecule and introduced an additional chirality correction (described in section 3.1.4).

Inference for all models variants was performed using 300 diffusion steps with a batch size of 128.

### 4.2. GEOM-DRUGS

We report results on recreating conformations from the GEOM-DRUGS dataset, compared with relevant baseline models, in Table 1. The dataset and benchmark are described in section 2.3. Table 1 is divided into sections based on categories of approximate model size, small (8-13M parameters), medium (25-64M parameters) and large (only MCF-L at 242M parameters), with columns indicating the number of conformers per molecule used during training ('$K_G$') and whether chirality information was used during training or generation ('Chirality'), to attempt to highlight the impact of various decisions made during model feature selection, data processing and training across previous models.

Our first model configuration, S23D-S-1/9 (S), shows a substantial improvement in quality of generated conformers relative to MCF-S, indicated by an improvement in all RMSD metrics, despite being $\approx 33\%$ smaller, thereby establishing a new SOTA for a small, non-equivariant MCG model. Within S23D-S variants, we observed similar performance from all coordinate encoding methods tested. Plain coordinate (C) and MLP (M) encodings gave a $\approx 1\%$ improvement in mean coverage for both recall and precision, compared with sinusoidal (S) encoding, but other metrics, such as median recall coverage, were not improved, so we left sinusoidal encoding as the default for further variant testing. A small improvement in some metrics seen when using 30 conformers per molecule rather than 20. Altering the ratio of graph encoder to structural processing blocks from 1/9 to 4/6 resulted in similar recall metrics but showed a noticeable improvement in all precision metrics.

S23D-B, our 'base' size model with 24.8M parameters, shows improved quality of generated conformers relative to MCF-B in all recall RMSD metrics despite being 61% smaller. In fact, S23D-B-1/13 (S) is within $0.2\%$ recall coverage of MCF-L, and close on other recall metrics. MCF-B performs better in precision metrics, but, as S23D-S improved on the precision metrics of MCF-S with less of a model size differential, we estimate that further scaling of the S23D-B model will surpass precision metric performance of MCF-B before reaching the same model size.

### 4.2.1. IMPACT OF CHIRALITY CORRECTION

Small model SOTA for recall AMR and precision coverage and AMR is set by ET-Flow, which used both chirality information as features and a chirality correction operation on generated conformers, inverting the molecule along one axis if incorrect chirality was detected after generation (Hassan et al., 2024). The ET-Flow - *SO(3)* variant modified the architecture to improve chirality matching, but removed the chirality correction step, which resulted in lower performance on RMSD metrics than ET-Flow with chirality correction, but still surpassed MCF-S in recall AMR as well as precision metrics. S23D-S variants show greatly improved recall coverage relative to ET-Flow variants, and S23D-S-1/9 (S) shows improved recall AMR when trained with 30 conformations per molecule, despite not using any chirality information during training or generation. Applying our simple chirality correction to the results from our S23D-S-1/9 (S) model trained with 20 or 30 conformers produced a $> 2\%$ increase in recall coverage and $\approx 6\%$ increase in precision coverage metrics. However, the equivariant ET-Flow model remains SOTA for precision, whether using chirality features or chirality correction. Note that this model even vastly outperforms the 242M parameter MCF-L model on precision metrics,

and used additional augmentations such as a harmonic prior and rotational alignment at the start of coordinate generation. Nevertheless, our best 'small' model produced a new SOTA for recall coverage, with mean $83.8\%$ and median $92.0\%$, for any model previously trained under $64M$ parameters.

When applying simple chirality correction to our 'base' size model, our S23D-B-1/13 (S) variant achieved a new SOTA for recall coverage on GEOM-DRUGS, with a mean of $86.5\%$, outperforming the MCF-L results with a model $\approx 10\%$ of the size. Mean recall AMR is virtually identical to that of MCF-L, and precision coverage and mean AMR are improved over MCF-L and only behind one set of results, the ET-Flow variant that used both chirality features and a chirality correction step.

## 5. Conclusions

In this paper, we introduced a new transformer model for MCG based on a standard transformer architecture combined with a new graph PE. The PE is represented as attention with a bias term that is linear with respect to the shortest path distance between atoms in a molecular graph. Our model demonstrates superior performance across multiple model sizes on a model-to-model basis. Two-stage training on molecules without hydrogen representation, followed by finetuning on complete molecules, will facilitate training of larger models on large datasets within a limited computational budget. This transformer can become the basis of large foundation models for accurate prediction of molecular structure. Thanks to recent advancements in AI, we foresee the emergence of large and computationally efficient models for fast MCG.

## 6. Limitations and Future Work

In the current model we did not use chirality information to construct the model features, which we view as the main limitation of this work and the immediate next direction of our research. We anticipate that providing the model the information required to understand stereochemistry will lead to a quantitative leap in accuracy and efficiency of MCG. Current splits used in the GEOM benchmarks are random and not based on molecule scaffolds, which could lead to limited real-world application even for models that achieve SOTA on these benchmarks (Guo et al., 2024). We have prepared new splits for our next study using Butina clastering (Hernández-Hernández & Ballester, 2023) on Murcko scaffolds (Wu et al., 2018) to test model generalization ability. Additionally, to speed up MCG for practical applications it is critical to explore how model quantization and distillation would affect the current results.

## 7. Impact Statement

In this paper, we advanced architectures of diffusion models for molecule structure generation. The architecture can be employed in pipelines for molecular property predictions, drug discovery, and lead optimization. Our work has a potential impact on several key areas of molecular modeling and brings AI closer to becoming a valuable tool across pharmaceutical research, materials science, and computational chemistry. Potential societal implications of this research arise from the use of AI tools in the advancement of drug discovery and the field of chemistry in general. Ethical concerns about this direction may include a lack of transparency in how molecule properties are calculated and the potential for misunderstanding of the results generated by AI models.

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

# Appendix

# A. Model details

## A.1. Model and training parameters

We trained the model with Adam optimizer with default PyTorch parameters and a constant learning rate 0.001, weight decay of 0.001 and batch size of 128. We trained the model on 2 V100 GPUs. GTX1660 and Quadro M5000 GPUs were used for inference.

*Table A1.* Transformer parameters

| Parameter | S23D-S-1/9 | S23D-S-4/6 | S23D-B-1/13 |
|---|---|---|---|
| Number of graph encoder blocks | 1 | 4 | 1 |
| Number of blocks in structure subnetwork | 9 | 6 | 13 |
| Hidden Size | 256 | 256 | 384 |
| Number of Attention Heads | 8 | 8 | 12 |

## A.2. Diffusion model

Following (Song et al., 2020b), we used stochastic differential equation (SDE)

$$dx = f(t)x dt + g(t)dw, \tag{9}$$

with initial condition $x(0) \sim p_0$ and $t \in (0, 1)$, where $w$ is a standard Wiener process, term $f(t)x$ is a drift term and $g(t)$ is a diffusion term. Here $x$ is a vector of stacked atom coordinates normalized by the constant 20Å. For variance preserving SDE (VPSDE), $f(t) = -\frac{\beta(t)}{2}$, $g(t) = \sqrt{\beta(t)}$, and $p(x(t)|x(0)) = \mathcal{N}(x(t); x(0)e^{-\frac{1}{2}\gamma(t)}, I - Ie^{-\gamma(t)})$, where $\gamma(t) = \int_0^t \beta(s)\,ds$. We used linear $\beta(t) = (\beta_e - \beta_s)t + \beta_s$ with $\beta$ in $(0.0, 18.0)$.

$$\gamma(t) = \int_0^t [(\beta_e - \beta_s)t + \beta_s]\,ds =$$
$$= (\beta_e - \beta_s)\frac{t^2}{2} + \beta_s t \tag{10}$$

The general formula to obtain noised samples in diffusion models is $x(t) = \alpha(t)x(0) + \sigma(t)\eta$, with $\alpha(t) = e^{-\frac{1}{2}\gamma(t)}$ and $\sigma(t) = \sqrt{1 - e^{-\gamma(t)}}$ for VPSDE. The noise schedule is highly important for the final model performance, as in (Chen, 2023) we modified the noise schedule

$$x(t) = \alpha(t)bx(0) + \sigma(t)\eta, \tag{11}$$

effectively rescaling atom coordinates by the constant $b = 20$, and the input to the score network was scaled by $b_s(t) = \frac{1}{\sqrt{(b^2-1)e^{-\gamma(t)}+1}}$ to keep stable network input variance. To train the score transformer $s_\theta(x, G, t)$, we used loss

$$\theta^* = \arg\min_\theta \mathbb{E}_t \left[ \mathbb{E}_{G,x(0)} \mathbb{E}_{x(t)|G,x(0)} \mathbb{E}_\eta \frac{1}{|V(G)|} ||s_\theta(x(t) \cdot b_s(t), G, t) \cdot \sigma(t) + \eta||_2^2 \right],$$

$$\tag{12}$$

where $G$ is a 2D molecular graph with number of atoms $|V(G)|$. The Euler–Maruyama method for reversed process was used to sample atom coordinates.

For molecules with size $M$, we worked in subspaces $X = \{x \in \mathbb{R}^{M \times 3} : \frac{1}{M}\sum_{i=1}^M x_i = 0\}$ by subtracting center of mass (CoM) from noise after sampling as shown in (Bao et al., 2022). We also subtracted the mean value from the output of the score network.

Multiple time tokens, each with a different embedding, were inputs to the score network. We used 4 time tokens instead of a single token to exploit the benefits of memory tokens (Burtsev et al., 2020). The projection of diffusion time encoding $TE$ was added to the time token embeddings,

$$
\begin{aligned}
TE(t, 2i) &= \sin(2^i \pi t) \\
TE(t, 2i+1) &= \cos(2^i \pi t)
\end{aligned}
\tag{13}
$$

where $i = 1, \ldots, 10$.

# B. Additional experiments

## B.1. GEOM datasets

### B.1.1. GEOM-QM9

We trained the model on the GEOM-QM9 dataset (Axelrod & Gómez-Bombarelli, 2022), introduced in section 2.3, using the data processed and provided by (Wang et al., 2024), see https://github.com/apple/ml-mcf. The processed test dataset contained 995 molecules rather than the 1000 indexed in the test set of (Ganea et al., 2021), so we used 995 for comparison with the MCF model, assuming that they tested on the molecules provided in their processed dataset. We trained for 430 epochs with batch size 512. Inference was performed with 300 diffusion steps, as in the GEOM-DRUGS experiments. In our data augmentation strategy for QM9 data, we switched from orthogonal transformations from O(3) group to rotations from SO(3) due to bias in the distribution of chirality in the QM9 dataset.

*Table A2.* Molecule conformer generation results on GEOM-QM9 ($\delta = 0.5$Å). Columns described in Table 1

| | SIZE (M) | $K_G$ | CHIRALITY | RECALL COVERAGE ↑ MEAN | MEDIAN | AMR ↓ MEAN | MEDIAN | PRECISION COVERAGE ↑ MEAN | MEDIAN | AMR ↓ MEAN | MEDIAN |
|---|---|---|---|---|---|---|---|---|---|---|---|
| GEOMOL | | 10 | YES | 91.5 | **100.0** | 0.225 | 0.193 | 87.6 | **100.0** | 0.270 | 0.241 |
| TOR. DIFF. | | 30 | YES | 92.8 | **100.0** | 0.178 | 0.147 | 92.7 | **100.0** | 0.221 | 0.195 |
| MCF-B | 64 | 10 | NO | 95.0 | **100.0** | 0.103 | 0.044 | 93.7 | **100.0** | 0.119 | 0.055 |
| ET-FLOW | 8.3 | 30 | YES | **96.5** | **100.0** | **0.073** | 0.047 | **94.1** | **100.0** | **0.098** | **0.039** |
| ET-FLOW - *SO(3)* | 9.1 | 30 | YES | 96.0 | **100.0** | 0.076 | **0.030** | 92.1 | **100.0** | 0.110 | 0.047 |
| S23D-B-1/13 (S) | 24.8 | 30 | NO | 95.6 | **100.0** | 0.097 | 0.048 | 91.2 | **100.0** | 0.145 | 0.070 |
| S23D-B-1/13 (S) | 24.8 | 30 | YES | 96.0 | **100.0** | 0.090 | 0.047 | 93.8 | **100.0** | 0.111 | 0.059 |

### B.1.2. GEOM-XL

As in other MCG papers (Jing et al., 2022; Wang et al., 2024; Hassan et al., 2024), we report results on GEOM-XL dataset, a subset of GEOM-MoleculeNet, to demonstrate how the model can generalize on large unseen molecules. For S23D-B-1/13 (S), we observed results comparable with MCF-B.

*Table A3.* Molecule conformer generation results on GEOM-XL. Columns described in Table 1

| | SIZE(M) | $K_G$ | CHIRALITY | RECALL AMR ↓ MEAN | MEDIAN | PRECISION AMR ↓ MEAN | MEDIAN |
|---|---|---|---|---|---|---|---|
| GEOMOL | | 20 | YES | 2.47 | 2.39 | 3.30 | 3.14 |
| TOR. DIFF. | | 30 | YES | 2.05 | 1.86 | **2.94** | 2.78 |
| MCF-S | 13 | 20 | NO | 2.22 | 1.97 | 3.17 | 2.81 |
| MCF-B | 64 | 20 | NO | 2.01 | 1.70 | 3.03 | 2.64 |
| MCF-L | 242 | 20 | NO | **1.97** | **1.60** | **2.94** | **2.43** |
| ET-FLOW | 8.3 | 30 | YES | 2.31 | 1.93 | 3.31 | 2.84 |
| S23D-B-1/13 (S) | 25 | 20 | NO | 2.07 | 1.80 | 3.22 | 2.83 |

### B.1.3. GEOM-DRUGS

To understand performance of our S23D-S-1/9 (S) and S23D-B-1/13 (S) model on the GEOM-DRUGS test set in more detail, we performed examined the RMSD metrics for recall and precision as a function of the coverage threshold (Fig.

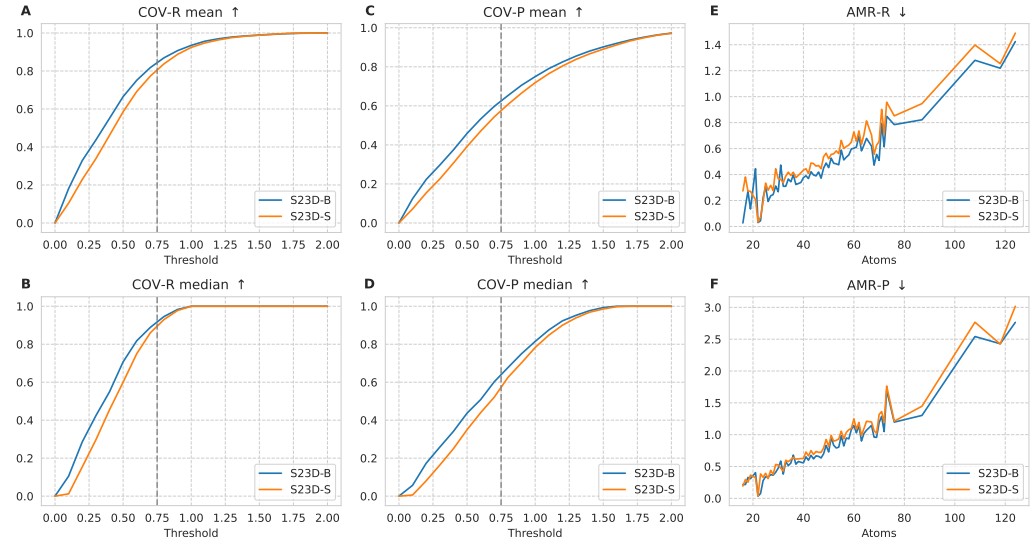

*Figure A1.* (**A-B**) Recall and (**C-D**) precision coverage as a function of coverage threshold, and **E** recall and **F** precision AMR as a function of number of atoms in a test molecule, for S23D-S-1/9 (S) and S23D-B-1/13 (S) (without chirality correction).

A1A-D) and number of atoms in the test molecules (Fig. A1E-F). This breakdown can be compared with Figure 4 in (Wang et al., 2024) and Figure 3 in (Hassan et al., 2024) to examine performance of S23D models against Torsional Diffusion, MCF and ET-Flow.

### B.2. Inference sampling

#### B.2.1. SAMPLING TIME

To test inference speed, we measured time per step during the diffusion process. We created batches from the GEOM-DRUGS validation split using different numbers of maximum atoms in each batch, and ran 1,000 steps of the diffusion process for the 'small' models S23D-S and MCF-S on both V100 and A100 GPUs. MCF models effectively double the batch size during inference to produce 2 conformations per input conformation, so we used batch size 128 in MCF-S and 256 in S23D-S. Results are shown in Fig. A2. On a V100 GPU, S23D-S is $\approx 4$ times faster per inference step than MCF-S for smaller molecules, and $\approx 2$ times faster for larger molecules. On an A100 GPU, S23D-S is $\approx 3$ times faster per inference step than MCF-S for smaller molecules, and slightly slower for larger molecules. Our architecture uses less computations than PerceiverIO (used in MCF), and is faster on devices preceeding the Ampere architecture, such as the V100, that do not support Flash Attention. MCF used a latent array in PerceiverIO of size 128, which is greater than the number of atoms in most molecules in the GEOM-DRUGS dataset. Thus, the latent layers of PerceiverIO have to process more "tokens" than our model, but this effect is reduced as molecule size approaches the latent size of 128. Currently Flash Attention doesn't support custom masks, so for S23D-S we are unable to benefit from the speed increase provided by Flash Attention on chips like the A100. In contrast, MCF doesn't require any mask in the latent layers and benefits from the Flash Attention optimization.

It is important to note that the mean number of atoms per molecule in GEOM-DRUGS is 44.4 (Axelrod & Gómez-Bombarelli, 2022), with s.d. 11.3, so the point at which MCF-S becomes faster than S23D-S on A100 ($> 70$ atoms per molecule) is rare in the dataset. Also, note that MCF performance metrics are reported using 1,000 inference steps, compared with 300 for S23D-S, making S23D-S $\approx 6$ times faster at the average molecule size to obtain the metrics reported in Table 1. We did not run timing comparisons against ET-Flow, but Figure 4 in (Hassan et al., 2024) demonstrates that ET-Flow is substantially slower per inference step than both MCF-S and MCF-B, which S23D models outperform.

#### B.2.2. INFERENCE STEPS

To demonstrate performance for different number of iterations during inference, we tested S23D-B-1/13 (S) with DDIM sampling (Song et al., 2020a). Results are shown in Table A4 using between 5 and 50 steps, compared with 300 in our

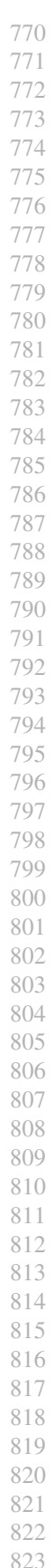

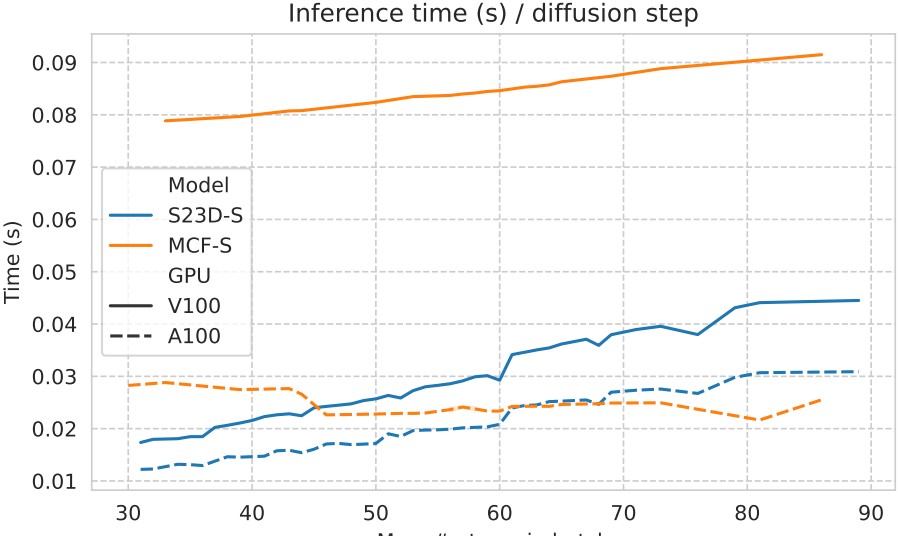

*Figure A2.* Inference time per diffusion step for S23D-S-1/9 (S) and MCF-S models on V100 and A100 GPUs over maximum number of atoms in a batch of size 256.

default implementation. No chirality corrections were used. We observed a modest drop in metrics, when the number of sampling iterations with DDIM were reduced to 50 and 20 iterations. However, when compared with the results in Table 1, S23D-B outperforms MCF-S on recall metrics using only 10 iteration steps with DDIM.

*Table A4.* Molecule conformer generation results on GEOM-DRUGS ($\delta = 0.75$Å) using different numbers of inference steps during sampling. Method shows inference method. Steps shows number of inference steps.

| | | | RECALL | | | | PRECISION | | | |
| | | | COVERAGE ↑ | | AMR ↓ | | COVERAGE ↑ | | AMR ↓ | |
| | METHOD | STEPS | MEAN | MEDIAN | MEAN | MEDIAN | MEAN | MEDIAN | MEAN | MEDIAN |
|---|---|---|---|---|---|---|---|---|---|---|
| S23D-B-1/13 (S) | EULER–MARUYAMA | 300 | 84.5 | 91.7 | 0.420 | 0.387 | 62.5 | 63.4 | 0.690 | 0.626 |
| S23D-B-1/13 (S) | DDIM | 50 | 83.0 | 91.3 | 0.447 | 0.422 | 58.8 | 59.4 | 0.734 | 0.676 |
| S23D-B-1/13 (S) | DDIM | 20 | 82.4 | 90.6 | 0.456 | 0.424 | 57.4 | 56.9 | 0.752 | 0.700 |
| S23D-B-1/13 (S) | DDIM | 10 | 81.0 | 89.7 | 0.477 | 0.450 | 55.5 | 54.9 | 0.776 | 0.723 |
| S23D-B-1/13 (S) | DDIM | 5 | 76.6 | 84.8 | 0.530 | 0.508 | 51.2 | 50.0 | 0.832 | 0.784 |

