# OpenReview forum: "A standard transformer and attention with linear biases for molecular conformer generation"
_ICML.cc/2025/Conference — Submitted to ICML 2025_

### Official Review · Reviewer_cbQ6 · 2025-03-09

**Overall Recommendation:** 3

**Summary:**

In this work, the authors introduce a transformer architecture with a new positional encoding scheme and training method for molecular conformer generation (MCG), achieving comparable performance as MCF, a prevailing non-equivariant MCG architecture, using a fraction of MCF's number of parameters. The proposed improvements include (1) positional encodings take inspiration from ALiBi and are subtractive rather than additive, (2) various coordinate encoding methods, and (3) chirality tagging for fair comparison with previous methods.

**Claims And Evidence:**

**Strengths**

- This is a highly empirical paper, and the authors do a good job of showcasing their results with comprehensive experiments.
- The authors cite efficiency as a highlight of MCF, and the sampling latency sudy in Appendix B gives good evidence of the efficacy of their

**Weaknesses**

- There is no analysis of the training efficiency of the proposed method versus baselines.
- While the proposed method outperforms other non-equivariant MCG architectures in terms of inference speed per diffusion step, there is no analysis on the convergence rate of the method versus other methods. While all models are evaluated with 300 diffusion steps, it's possible certain models achieve high-quality samples sooner than others. Measuring metrics against diffusion steps would be helpful in more comprehensively evaluating efficiency.

**Essential References Not Discussed:**

This work is relatively focused in the research problem it is solving. To my knowledge, the authors discuss the most relevant references.

**Experimental Designs Or Analyses:**

- What are the details regarding data augmentation (number of augmentations per sample, any post-processing or filtering, etc.)?
- I found no core issues with the experimental design for Section 4.

**Methods And Evaluation Criteria:**

**Strengths**

- The benchmark datasets and metrics are all standard in the MCG literature.

**Weaknesses**

- It's unclear why the attentional linear biases are subtractive rather than additive as they are for Graphormer and other transformer variants with linear biases.
- Are competing baselines also trained using data augmentation? To my knowledge, MCF doesn't use any data augmentation, which could weaken the efficiency of the proposed method.

**Other Comments Or Suggestions:**

Some typos:

- "_non-equivariant models_ can outperform _non-equivariant_ networks at MCG"
- "_stereochemestry_"
- "ET-Flow, _utlizing_ an equivariant transformer"
- “Inference for all _models_ variants"

**Other Strengths And Weaknesses:**

The proposed method showcases comparable non-equivariant MCG performance with much fewer parameters than MCF. While the claims made are relatively modest, with the paper's main contributions being what is essentially a thorough search over architecture choices, I think it marks good progress towards more efficient and simpler methods for a complicated task.

**Questions For Authors:**

Please address the weaknesses and questions posed in the preceding sections.

[1] Jing et al. Torsional Diffusion for Molecular Conformer Generation. NeurIPS 2022.
[2] Wang et al. Swallowing the Bitter Pill: Simplified Scalable Conformer Generation. ICML 2024.
[3] Qu, Eric and Krishnapriyan, Aditi S. The Importance of Being Scalable: Improving the Speed and Accuracy of Neural Network Interatomic Potentials Across Chemical Domains. NeurIPS 2024.
[4] Brehmer et al. Does equivariance matter at scale? Preprint.

**Relation To Broader Scientific Literature:**

This paper falls in line with recent work on non-equviariant MCG models [2] and scaling studies for molecular and geometric models [3, 4].

[3] Qu, Eric and Krishnapriyan, Aditi S. The Importance of Being Scalable: Improving the Speed and Accuracy of Neural Network Interatomic Potentials Across Chemical Domains. NeurIPS 2024.

[4] Brehmer et al. Does equivariance matter at scale? Preprint.

**Theoretical Claims:**

The paper makes no theoretical claims.

---

> ### Author Rebuttal · Authors · 2025-04-01
>
> Thank you to the reviewer for the insightful comments. We have performed additional analyses, which we hope will address the reviewer's concerns.
>
> > *"...no analysis of the training efficiency of the proposed method versus baselines"*
>
> Training time should be roughly proportional to single step inference time provided in Figure A2.
>
> Additionally, in the response to Reviewer qodJ we provide detail from our two-stage training in the form of the metrics achieved by model variants after 100 training epochs with no hydrogens.
>
> > *"Measuring metrics against diffusion steps would be helpful..."*
>
> The number of diffusion steps to achieve high quality samples depends on the type of diffusion or flow-matching model. We chose a diffusion model to make comparison with MCF, but in our future work we anticipate that our model will work well with flow matching and a harmonic prior, as implemented in ET-Flow, and would require fewer iterations.
>
> Nevertheless, we demonstrated DDIM sampling using S23D-B-1/13 (S) in Table A4 of the Appendix. At 50 steps we observed only a small drop in metrics compared with 300 with default sampling. In terms of convergence rate compared with the MCF model, we noted that at 10 DDIM steps our 25M parameter model outperforms the 13M MCF model, which used 1000 diffusion steps. DDIM sampling for MCF was reported in Figure 6 of the MCF paper, showing approximately comparable results with our Table A4 between the 64M MCF-B model and 25M S23D-B model, both surpassing 80% COV-R between 5-10 DDIM steps. The MCF-B result was recapitulated in Figure 4 of the ET-Flow paper, where they also demonstrate that ET-Flow does not suffer from reducing the number of steps and can use just 5 steps for inference.
>
> To compare convergence rate across different methods choices for the S23D-S model, we have performed new inference runs using Euler–Maruyama at 50, 100, and 200 inference steps for S23D-S, -M, and -C variants. The results, shown in the table below, indicate that performance is similar at 300, 200, or 100 inference steps, and drops by a similar amount for all metrics across all model variants at 50 steps. Our initial choice to report results at 300 steps was cautious, and we could have obtained similar results using 100-200.
>
> |Model|Steps|Recall||||Precision||||
> |-|-|-|-|-|-|-|-|-|-|
> |||COV||AMR||COV||AMR
> |||Mean|Med|Mean|Med|Mean|Med|Mean|Med
> |S23D-S-1/9 (S)|300|80.7|89.9|0.483|0.458|57.5|57.5|0.757|0.700
> ||200|81.1|89.8|0.483|0.455|57.7|57.1|0.755|0.701
> ||100|80.9|88.9|0.495|0.470|57.4|56.7|0.767|0.720
> ||50|78.9|87.1|0.551|0.517|54.8|53.9|0.817|0.765
> |S23D-S-1/9 (M)|300|81.5|90.0|0.473|0.444|58.1|57.8|0.751|0.702
> ||200|81.7|89.5|0.476|0.451|57.6|56.9|0.753|0.706
> ||100|81.4|89.3|0.488|0.461|57.2|56.3|0.764|0.713
> ||50|79.2|87.1|0.546|0.518|54.9|53.1|0.817|0.758
> |S23D-S-1/9 (P)|300|81.9|88.9|0.462|0.442|58.8|58.4|0.740|0.697
> ||200|81.2|88.3|0.474|0.445|57.8|57.2|0.753|0.703
> ||100|81.5|88.9|0.485|0.457|58.2|57.0|0.759|0.702
> ||50|79.2|86.7|0.546|0.517|55.6|54.6|0.809|0.743
>
> > *"It's unclear why the attentional linear biases are subtractive..."*
>
> Graphormer uses additive bias because the bias is learnable and any sign could be used. However, we configured the bias to decrease attention weights with the distance between nodes, which is likely to be what is learned by Graphormer.
>
> > *"Are competing baselines also trained using data augmentation?"*
>
> MCF did test the impact of data augmentation (see Figure 5 of (Wang et al., 2024)), but found augmentation applying a different rotation to each conformer during the training epoch (our augmentation strategy, labelled "Random" in their Figure 5) to be highly detrimental due to a bias in the dataset. This could indicate that we are, in fact, reducing the performance of our method on the current benchmark by including data augmentation. However, augmentation is critical for better model generalization on other datasets. Also, in the MCF code they include a fixed non-symmetric normalization of coordinates for the three different axes, which could worsen performance if a dataset changes. For instance for GEOM-DRUGS, they use a MinMax scaler with min_x=-16.8, max_x=16.6, min_y=-10.5, max_y=10.7, min_z=-7.1, max_z=7.4.
>
> > *"What are the details regarding data augmentation (number of augmentations per sample, any post-processing or filtering, etc.)?"*
>
> We did online augmentation for every sample in each batch. We used the same dataset from the MCF study, with no post-processing or filtering. To calculate RMSD we used the Posebusters package which calls the "getBestRMS" function from RDKit.
>
> > *"...recent work on non-equviariant MCG models [2] and scaling studies for molecular and geometric models [3, 4]"*
>
> We will add references Qu & Krishnapriyan, 2024 and Brehmer et al., 2024.

---

> > ### Comment · Reviewer_cbQ6 · 2025-04-04
> >
> > Thank you for the response. I am overall satisfied with the replies, especially with the study on the impact of the number of steps on performance. I retain my assessment of this work and my rating.

---

> > > ### Author Response · Authors · 2025-04-07
> > >
> > > We thank the reviewers for their valuable feedback. We hope that our clarifications and the additional analyses and ablation experiments suggested by all the reviewers significantly strengthen the final manuscript.

---

### Official Review · Reviewer_kw5E · 2025-03-09

**Overall Recommendation:** 3

**Summary:**

This paper presents "S23D", a molecular conformer generation approach based on a standard transformer augmented with linear attention biases (similar to ALiBi). Unlike specialized equivariant models, S23D relies on graph-relative positional encodings and achieves competitive results at smaller parameter counts. The authors demonstrate state-of-the-art (SOTA) performance for recall metrics on the GEOM-DRUGS benchmark with a relatively small transformer (25M parameters vs. 64M in prior models). Key ideas include relative positional encoding using shortest-path distances as attention biases, a two-stage training strategy (first without hydrogen atoms, then with hydrogens reintroduced), and a post-hoc chirality correction step.

**Claims And Evidence:**

Main claims—improved performance with smaller transformer due to positional encoding—are only partially supported. While S23D slightly surpasses MCF-B (64M parameters) in recall metrics (84.5% vs 84.0% mean coverage), precision metrics lag slightly behind without additional chirality handling (Table 1). The paper suggests this encoding alone addresses size limitations, but lacks direct ablation experiments clearly isolating positional encoding contributions from other differences (e.g., transformer backbone, training setup).

**Essential References Not Discussed:**

Several important prior works are not cited or discussed adequately:
- Molecular Geometry Prediction Using a Deep Generative Graph Neural Network [Mansimov 2019]
- Symmetry-adapted generation of 3d point sets for the targeted discovery of molecules[Gebauer et al. 2019]
- Classical approaches (OMEGA, RDKit ETKDG) performance on GEOM benchmarks, which provide useful performance baselines.

**Experimental Designs Or Analyses:**

Experiments are comprehensive, clearly comparing S23D variants to prior methods (GeoMol, MCF, ET-Flow) using established GEOM splits and metrics (Table 1). However, the evaluation lacks explicit comparisons with classical baseline methods (e.g., OMEGA, RDKit) to contextualize gains clearly. Also missing is deeper qualitative or error-mode analysis—e.g., identifying specific molecules or conformations where the proposed method struggles.

**Methods And Evaluation Criteria:**

Methods are appropriate and standard for molecular conformer generation: a diffusion-based generative model using GEOM-DRUGS, and standard RMSD-based coverage metrics (COV, AMR). The attention bias approach (ALiBi-like linear bias on shortest-path distances) is reasonable but not fully justified; no rigorous ablation is done to confirm its necessity or optimality compared to simpler encodings. Additionally, statistical rigor (e.g., multiple seeds, confidence intervals) is not provided, leaving uncertainty around minor metric improvements reported in Table 1.

**Other Comments Or Suggestions:**

Minor issues:
- several typos (e.g., "seperated" → "separated" in Table 1 caption, "utlizing" → "utilizing").
- A direct ablation study removing positional bias entirely would strongly improve the paper's clarity and rigor.

**Other Strengths And Weaknesses:**

Strengths:
- Clearly organized experiments, and use of standard metrics and splits
- reasonable model design (transformer with relative bias).

Weaknesses:
- Incremental novelty (combination of existing known techniques without deep novelty
- Superficial theoretical justification
- Lack of clear ablations (e.g., importance of positional bias) with limited qualitative/error analysis.

Although the paper reports good empirical results, particularly regarding smaller model size, the contributions are primarily incremental—combining known techniques without deep novelty or rigorous theoretical justification. The lack of essential ablation studies, incomplete discussion of prior foundational work, and limited qualitative analysis reduce its significance. The work is solid empirically but insufficiently novel or insightful to clearly justify acceptance at ICML.

**Questions For Authors:**

- Have you conducted direct ablations showing performance without any positional biases? How critical is the ALiBi-style attention bias?
- How would explicitly adding chirality information during training (as ET-Flow did) affect your results? Could it address multi-center chirality better?
- What are the limitations regarding molecule size (due to quadratic attention cost)? Did you encounter computational or scaling issues with larger molecules?
- Have you assessed generalization beyond GEOM-DRUGS (e.g., natural products, macrocycles)? Does your method generalize or require retraining?

**Relation To Broader Scientific Literature:**

The paper adequately cites key recent works on molecular conformer generation (MCF, ET-Flow, Torsional Diffusion) and graph transformers (Graphormer, ALiBi). However, important foundational references like [Molecular Geometry Prediction Using a Deep Generative Graph Neural Network, Mansimov 2019] or early generative works (G-SchNet, CVGAE) are omitted, which are relevant for context.

**Theoretical Claims:**

The paper does not contain explicit theoretical claims or proofs. Implicit assumptions about the sufficiency of rotational augmentation (to compensate for lack of equivariance) and linear bias effectiveness are heuristic. The theoretical grounding of why linear bias specifically improves transformer performance is not provided, limiting theoretical insights.

---

> ### Author Rebuttal · Authors · 2025-04-01
>
> We sincerely thank the reviewer for their constructive critique. We made additional analyses including ablation experiments that will be added to the manuscript to address the comments.
>
> We made 3 runs with ablation on PE for the S23D-S-1/9 (S) model using 1-stage training (with hydrogen) for 100 epochs (no chirality correction):
>
> 1. ALiBi PE
> 2. MCF Eigenvector (EV) encodings, added as new features in our Eq. 5; k=28 as in MCF (28 vectors)
> 3. PE used in Graphormer, using our Eq. 1 updated for multiple heads as in Graphormer (with max. shortest path threshold of 20, as in the default Graphormer settings)
>
> |Model|Recall||||Precision||||
> |-|-|-|-|-|-|-|-|-|
> || COV||AMR||COV||AMR||
> ||Mean|Med|Mean|Med|Mean|Med|Mean|Med|
> |S23D-S-1/9 (ALiBi)|81.4|89.0|0.472|0.449|57.5|57.0|0.756|0.709|
> |S23D-S-1/9 (EV) |79.5|86.9|0.507|0.479|55.6|54.5|0.789|0.739|
> |S23D-S-1/9 (Learnable Bias)|81.5|88.9|0.468|0.441|58.6|59.5|0.744|0.693|
>
> Run 1 (default run with S23D-S-1/9 (S) with hydrogens) was required to provide a new baseline for the other ablation runs, as the eigenvectors we used from the MCF dataset were computed for graphs with hydrogens. Run 2 (EV) recall results are consistent with the MCF-S model. In Run 3 results were comparable with our PE. However, the training was 35% slower than run 1 on A100 GPUs due to inefficient lookup operations, as discussed in our methods section. In the future, it is possible to explore a variant of our PE with learnable slopes for each head that would combine advantages of Graphormer PE and ALiBi. The training dynamics of mean COV-R for the 3 runs are shown in the table:
>
> |Model|Number of Epochs||||
> |-|-|-|-|-|
> ||25|50|75|100|
> |S23D-S-1/9 (ALiBi)|75.4|79.4|80.0|81.4|
> |S23D-S-1/9 (EV) |70.3|75.9|79.3|79.5|
> |S23D-S-1/9 (Learnable Bias)|77.0|80.5|81.4|81.5|
>
> Additionally, we trained the base model for 85 epochs (with hydrogens) using the eigenvector encoding scheme, which roughly corresponds to our two-stage training (see discussion in response to reviewer qodJ), which gave worse metric results than using ALiBi:
>
> |Model|Recall||||Precision||||
> |-|-|-|-|-|-|-|-|-|
> ||COV||AMR||COV||AMR||
> ||Mean|Med|Mean|Med|Mean|Med|Mean|Med|
> |S23D-B-1/13 (EV)|82.7|90.0|0.449|0.422|60.0|60.1|0.725|0.664|
>
> We will add these PE ablation results into the experiments section of the manuscript.
>
> > *"...prior works are not cited..."*
>
> We will add discussion of the Mansimov et al. 2019 (CVGAE) and Gebauer et al. 2019 (G-SchNet) references.
>
> > *"...explicit comparisons with classical baseline methods (e.g., OMEGA, RDKit)"*
>
> OMEGA and RDKit have been tested in this task on the GEOM datasets in previous publications, for example GeoMol and Torsional Diffusion, with repeated findings that these methods perform poorly at this task so we did not recreate the analysis here.
>
> > *"Have you conducted direct ablations showing performance without any positional biases? How critical is the ALiBi-style attention bias?"*
>
> We did not run ablations without any positional biases. The setup requires some form of PE to define edges of a molecular graph. We could use RDKit to reconstruct bonds, but there is no guarantee that resulting molecules will be the same as the source molecules, and it is not clear how to test results.
>
> > *"How would explicitly adding chirality information during training (as ET-Flow did) affect your results?"*
>
> After paper submission we made attempts to incorporate chirality features, but our model was not able to capture correct chiralities. Additionally, we checked conformations generated by ET-Flow and found that their model also produces conformations with incorrect multi-center chirality.
>
> > *"What are the limitations regarding molecule size (due to quadratic attention cost)?"*
>
> Quadratic complexity is not an issue for our model. Scaled dot product attention even without FlashAttention is highly optimized in PyTorch. That is one of the benefits of using a standard transformer.
>
> > *"Have you assessed generalization beyond GEOM-DRUGS..."*
>
> No, we did not. We anticipate that the model would require finetuning in this case.
>
> > *"...the contributions are primarily incremental—combining known techniques without deep novelty..."*
>
> We would argue against the reviewer’s description of the work as primarily incremental. Although the model combines known components, creating an illusion of simplicity, it was not trivial to create the simplest and fastest transformer model that successfully competes with equivariant counterparts at small model sizes. For instance, compare our model with the complex ET-Flow transformer or recent non-equivariant networks. The presented positional encoding is simple in implementation, efficient in computations and could be further optimized with Triton kernels. These factors are critical for practical model applications in MCG.

---

### Official Review · Reviewer_qodJ · 2025-03-12

**Overall Recommendation:** 4

**Summary:**

The paper introduces a novel relative positional encoding technique similar to the ALiBi technique found in NLP for non-equivariant Transformer diffusion models for generating molecular conformations. The proposed approach allows scaling down non-equivariant Transformer, which typically requires a large model size to compensate for the lack of equivariant bias. The authors show that the proposed method allows a standard Transformer with 25M parameters to outperform previous state-of-the-art non-equivariant models with 64M parameters on the GEOM-DRUGS benchmark.

**Claims And Evidence:**

The main contribution of the paper is the introduction of a relative positional encoding technique, which the authors claim allows for scaling down non-equivariant Transformer models for generating molecular conformations. The claim is supported by convincing evidence on the GEOM-DRUGS benchmark.

The additional claim that the proposed two-stage training protocol (without and with hydrogen atoms) "makes more efficient use of a limited computational budget when scaling models" is not supported by evidence. The authors provide the inference speed with respect to the number of atoms in the batch, but do not show the performance degradation or improvement resulting from the two-stage approach. While efficiency is improved, the performance degradation might be such that one would be better off training with the hydrogen atoms from the start. I would be convinced by a comparison of the performance of two identical models trained using and without the proposed two-stage approach, with the same number of FLOPS.

**Essential References Not Discussed:**

Not to my knowledge, although I am not familiar with the literature on molecular conformation generation.

**Experimental Designs Or Analyses:**

The experimental design seems fair and to take into account all essential experimental design that would influence the performance on the benchmark: the model size, the number of conformations considered, as well as the presence of a mechanism to address the chirality.

**Methods And Evaluation Criteria:**

The authors rely on the GEOM-DRUGS benchmark, following the same splits and metrics as previous works in the field. I believe the evaluation methodology is suitable for evaluating the quality of the conformation generated by the proposed approach, as it has previously been used by several previous works.

**Other Comments Or Suggestions:**

The left-side of Figure 1 was a bit confusing on the first pass. For instance, I did not understand where the coordinates on the middle-left came from, and I did not know what the "atom tokens" inputs were. I suggest you add additional arrows and explicitly write the input. You could also add light colors to better separate the different parts and mention in the caption that "The first transformer blocks (blue) encode...", "Coordinate encodings (orange) are projected...", etc.

**Other Strengths And Weaknesses:**

The introduction and related work sections are particularly well written and extensive. They provide a great introduction into the field of molecular conformation generation.

However, the methodology section could benefit from more details for readers who might not be familiar with the Variance Preserving SDE. I appreciate the description in the appendix, and the methodology doesn't need to be fully self-contained since it relies on an existing model. Still, I believe that providing more high-level details would improve the reader’s understanding. Notably, some of the design choices are not explained (or mentioned that they were taken from previous work). For instance, why only considering chemical elements with frequent occurrences in the vocabulary? Why masking 10% of atoms only for 1% of the molecules?

**Questions For Authors:**

- Can you provide a more comprehensive description of the models and design choices?
- Can you add the RMSD for the examples provided in Figure 1?

**Relation To Broader Scientific Literature:**

The paper makes a significant effort to introduce the relevant works and to position themselves in the broader scientific literature. They augment the existing Variance Preserving SDE diffusion model with their proposed relative positional encoding, and they highlight the importance of including the chirality information, which was previously known.

**Theoretical Claims:**

N/A

---

> ### Author Rebuttal · Authors · 2025-04-01
>
> We thank the reviewer for reviewing the manuscript and providing valuable comments. We have performed additional analyses, described below, in light of these comments, and hope that they alleviate the reviewer's concerns.
>
>  > *"I would be convinced by a comparison of the performance of two identical models trained using and without the proposed two-stage approach, with the same number of FLOPS."*
>
> We initially had two motivations for our two-stage training protocol, first to make efficient use of computational resources during early testing of multiple model configurations, and second to test the necessity of including hydrogen atoms during training of a model for MCG, due to the possibility of using fast, rule-based approaches to add hydrogens after generation of heavy atom coordinates (e.g., Chem.AddHs in RDKit). We achieved SOTA metric values after the first stage of training, showing that training without hydrogen is a viable approach. The primary motivation for the second stage of training, and the reason we reported those results in the manuscript, was for a direct comparison with previous methods, which all generate coordinates for all atoms, including hydrogens. After the second stage of training, metrics were similar to those found after the first stage. Unfortunately, we neglected to include the results after 100 epochs without hydrogens in the submitted version of the manuscript, but we will add those to the appendix and we report the values in the table below (no chirality correction):
>
> |Model|Recall||||Precision||||
> |-|-|-|-|-|-|-|-|-|
> || Coverage||AMR||Coverage||AMR||
> ||Mean|Median|Mean|Median|Mean|Median|Mean|Median|
> |S23D-S-1/9 (C)|80.8|88.7|0.485|0.454|57.5|57.1|0.763|0.707|
> |S23D-S-1/9 (S)|80.3|89.6|0.495|0.467|56.1|55.4|0.779|0.724|
> |S23D-S-1/9 (M)|81.0|89.4|0.485|0.457|58.2|59.1|0.751|0.702|
> |S23D-B-1/13 (S)|84.1|91.2|0.428|0.399|62.2|64.3|0.694|0.625|
> |S23D-B-1/13 (M)|84.2|91.9|0.425|0.391|61.7|63.2|0.701|0.632|
>
> To demonstrate efficiency of two-stage training, in an additional run we tested single-stage training with hydrogens at 50+35=85 epochs. Training time per epoch without hydrogens was approximately half of that with hydrogens, so the first 50 epochs of training with hydrogens is approximately equivalent to the first stage of training (100 epochs) without hydrogens. After 50 epochs of training with hydrogens, COV-R was 79.4, compared to 80.3 after 100 epochs without hydrogens.
>
> The additional 35 epochs represent the second stage of training with hydrogens. The model results we obtained at 85 epochs were approximately the same as in two stage training, and are shown in the table (no chirality correction):
>
> |Model|Recall||||Precision||||
> |-|-|-|-|-|-|-|-|-|
> || Coverage||AMR||Coverage||AMR||
> ||Mean|Median|Mean|Median|Mean|Median|Mean|Median|
> |S23D-S-1/9 (S)|81.1|88.9|0.481|0.459|58.8|58.3|0.744|0.699|
>
> However, the two-stage training approach allows the researcher to do faster testing of different architectures, to use commodity GPUs for training, and facilitates hyperparameter tuning. The table below shows COV-R for single-stage training at intervals during training with hydrogens for 100 epochs (no chirality correction):
>
> |Model|Number of Epochs||||
> |-|-|-|-|-|
> ||25|50|75|100|
> |S23D-S-1/9 (S)|75.4|79.4|80.0|81.4|
>
> > *"... the methodology section could benefit from more details for readers who might not be familiar with the Variance Preserving SDE"*
>
> We will add a paragraph to the Related Work section with an introduction to diffusion and flow matching models.
>
> > *"For instance, why only considering chemical elements with frequent occurrences in the vocabulary? Why masking 10% of atoms only for 1% of the molecules?"*
>
> The idea of using only frequent elements is taken from NLP, wherein rare tokens are not included in vocabularies and are replaced with the UNK token (here we used the MASK token). MASK is used to randomly replace other tokens, forcing the model to learn contextual representation. The probabilities 10% and 1% were chosen arbitrarily.
>
> > *"The left-side of Figure 1 was a bit confusing on the first pass. ... I suggest you add additional arrows and explicitly write the input."*
>
> We will update Figure 1 with a more detailed description of each transformer block.
>
> > *"Can you add the RMSD for the examples provided in Figure 1?"*
>
> RMSD values, from top to bottom, are 0.29, 0.85, 0.15. Note that we found an error in Figure 1. The images for the bottom molecule were extremely similar and we accidentally mixed the ground truth and generated images. We will correct this, and include RMSD values, in the revised manuscript.

---

> > ### Comment · Reviewer_qodJ · 2025-04-03
> >
> > I appreciate the reviewers' clarifications and am pleased with the additional experiments. I have updated my score accordingly.

---

> > > ### Author Response · Authors · 2025-04-07
> > >
> > > We would like to thank the reviewers again for their comments, which helped us to shape the manuscript by providing additional clarifications and experiments that we think greatly improve the work. Thank you for taking the time to go through the paper in such detail.

---

### Official Review · Reviewer_gswC · 2025-03-14

**Overall Recommendation:** 3

**Summary:**

In this paper the authors propose a new method for sampled-based molecular conformer generation that is built on top of a non-equivariant model. They show that unlike prior methods, with the right modifications to the architecture and the right training procedures, one can train non-equivariant architectures that do not require orders of magnitude more parameters than their equivariant counterparts.

Specifically, the paper proposes to use standard transformer blocks from LLaMA, first encoding the 2d graph, and then using 3d coordinates, themselves encoded with periodic encodings. The attention mechanism is biased so as to prefer attending to graph-nearby nodes. The model is then trained with a standard denoising diffusion objective to learn to approximate the score function. This training is split in two stage, where hydrogen atoms are only included in the structure in the second phase.

## update after rebuttal

Thanks for the rebuttal. I appreciate that the authors have run additional analyses to further provide evidence, specifically orthogonal evidence such as running PoseBusters. Answering the _why_ of a method is always more valuable that just its effects. As for scaling, I acknowledge that not all researchers possess the same amount of compute, but I'd remind us that scaling power laws go both ways: training smaller models is a valid way of obtaining scaling trends.

**Claims And Evidence:**

The 3 core claims are that the proposed model achieves good performance, that chirality matters in evaluation, and that the proposed two-stage scheme is compute efficient. These claims are mostly supported by evidence. There is a fourth broader claim motivating the paper, which is that the specific methods used here scale better than prior work. I'd argue that the evidence here is quite limited, and is in fact, limited to 2 data points (8.6M and 25M parameters). While this is obviously enough to get a sense of the claim, there is not enough to get a deeper understanding of the scaling behaviors of the different choices made here (e.g. only 1/13 (S) is tested at 25M parameters).

There is another unfortunate lack of results, in that all results are about performance. There is a richness in molecular structures that is underexplored here. Why are sinusoidal encodings helpful? What do they model? Do they _actually_ scale better (it seems like their use in the 25M parameter was simply extrapolated from their being a good choice in the 8.6M regime)? Perhaps these encodings are better able to capture bond lengths, or angles, or whatnot. These are things that could be measured and bring much more clarity to the paper. As is, we've learned very little about the modifications introduced in this paper other than they work in this specific regime for this specific model.

**Essential References Not Discussed:**

Nothing I can think of.

**Experimental Designs Or Analyses:**

See Claims section.

**Methods And Evaluation Criteria:**

The methods and evaluation are standard, and make sense here.

**Other Comments Or Suggestions:**

My main suggestion is really, as above, to do more to _teach_ the audience something about the work. Making numbers go higher is great, but what's even better is understanding more precisely what the mechanisms behind our methods are. This is the best way for science to progress.

**Other Strengths And Weaknesses:**

The paper is quite straightforward, well explained.

**Questions For Authors:**

Another major consideration in comparing to models like ETFlow is their very training method, i.e. flow matching. Do the authors have a sense of whether certain architectures are more amenable to flow matching than to diffusion training? Whether scaling behaviors would change at all?

**Relation To Broader Scientific Literature:**

I think the paper puts forward work that's important in the current context. The question very much remains open as to which class of model, equivariant or not, will end up being the most practical and end up scaling to the larger systems which one hopes these models will one day be applied. While this work by no means settles the debate, it is a welcome data point.

**Theoretical Claims:**

Not applicable.

---

> ### Author Rebuttal · Authors · 2025-04-01
>
> We thank the reviewer for valuable comments. To address the comments, we made additional analyses and hope our response will alleviate the reviewer’s concerns.
>
> > *"I'd argue that the evidence here is quite limited, and is in fact, limited to 2 data points (8.6M and 25M parameters). While this is obviously enough to get a sense of the claim, there is not enough to get a deeper understanding of the scaling behaviors of the different choices made here (e.g. only 1/13 (S) is tested at 25M parameters)."*
>
> We selected two datapoints because models with greater than 25M parameters present practical difficulties in MCG, and the computational resources required to train and test larger models (e.g., 64M or 242M as in MCF) are not accessible to many researchers. To further demonstrate scaling behavior with a different model choice, we made an additional training run with 25M parameters with the MLP coordinate encoder, no chirality correction:
>
> |Model|Recall||||Precision||||
> |-|-|-|-|-|-|-|-|-|
> || Coverage||AMR||Coverage||AMR||
> ||Mean|Median|Mean|Median|Mean|Median|Mean|Median|
> |S23D-B-1/13 (M)|84.6|91.9|0.412|0.381|62.4|64.1|0.684|0.613|
>
> > *"Perhaps these encodings are better able to capture bond lengths, or angles, or whatnot."*
>
> To study the effect of coordinate encodings on generated molecular structure, we ran additional tests from the PoseBusters package, including assessment of bond lengths, bond angles, aromatic ring flatness, planar double bonds, and internal steric clashes, on our generated molecules. Using the default criteria for "intramolecular validity" features in PoseBusters that a predicted feature (e.g. bond length) should be within +/- 25% of the reference for that feature, all 5 of the above features were perfectly captured, with >99.2% pass rate for all model variants that we reported in Table 1. Lowering the thresholds from 25% to 10% and 5% we saw metrics drop evenly across model variants. For instance, at 5% correct bond angles were found at a rate of 58.6% for S23D-S-1/9 (S), 56.4% for S23D-S-1/9 (M), and 54.8% for S23D-S-1/9 (C).
>
> > *"It's quite interesting to see the error scaling linearly as a function of the number of atoms, but I do wonder if this paints the right picture. I suspect that if one measured the energy of those conformers, we would instead see an exponential increase in energy as a function of the number of atoms."*
>
> We agree with the reviewer that error increases with molecule size. Considering energies and populations in the generated conformational ensemble is a topic we will investigate in future work.
>
> > *" ... the authors have a sense of whether certain architectures are more amenable to flow matching than to diffusion training? Whether scaling behaviors would change at all?"*
>
> We do not anticipate any particular transformer model having a unique advantage over others when applied to different architectures of diffusion or flow matching models. We used a diffusion model to compare our model with the MCF model, and the next step will be an update of the diffusion block to a more advanced architecture (e.g., Inductive Moment Matching).

---

### Decision · Program_Chairs · 2025-05-01

**Decision:**

Reject

**Comment:**

The paper incorporates relative positional encoding as a negative attention bias into the standard transformerfor molecular conformer generation (MCG). The authors claim that their model surpasses the current state-of-the-art non-equivariant base model on the GEOM-DRUGS benchmark with fewer parameters.

While there are three weak accept recommendations and one accept, the evaluation of the work is not solid and does not meet the conference's standards:
- main paper tests only one dataset. Although results for additional datasets are included in the Appendix, the proposed method performs worse than previous methods on these datasets.
- Some sota methods, such as DMCG, are not compared. "One of the first non-equiv models, DMCG (Zhu et al., 2022), led benchmarks on molecular conformation generation for a relatively long time and was succeeded by the Molecular Conformer Fields (MCF) model (Wang et al., 2024), which achieved groundbreaking root mean square deviation (RMSD) metrics on the GEOM-DRUGS benchmark." In the MCF paper, DMCG and MCF are not compared, so it remains unclear which of the two performs better. A comparison with DMCG is recommended.

Overall, the effectiveness of the proposed method is not well substantiated, and the empirical evaluation appears insufficient for ICML.